# Fragile neutrophils in surgical patients: A phenomenon associated with critical illness

**Lillian Hesselink**[1,2]*, **Roy Spijkerman**[1,2], **Pien Hellebrekers**[1,2], **Robert J. van Bourgondiën**[1], **Enja Blasse**[3], **Saskia Haitjema**[3], **Albert Huisman**[3], **Wouter W. van Solinge**[3], **Karlijn J. P. Van Wessem**[1], **Leo Koenderman**[2,4¤a], **Luke P. H. Leenen**[1], **Falco Hietbrink**[1¤b]*

**1** Department of trauma surgery, University Medical Center Utrecht, Utrecht, the Netherlands, **2** Center for Translational Immunology, Wilhelmina Children's Hospital, Utrecht, the Netherlands, **3** Department of Clinical Chemistry and Hematology, Utrecht, the Netherlands, **4** Department of Respiratory Medicine, University Medical Center Utrecht, Utrecht, the Netherlands

¤a Current address: Wilhelmina Children's Hospital, Utrecht, the Netherlands
¤b Current address: University Medical Center Utrecht, Utrecht, the Netherlands

* f.hietbrink@umcutrecht.nl (FH); l.hesselink@umcutrecht.nl (LH)

**Data Availability Statement:** Data cannot be shared publicly because of European General Data Protection Regulation (EU GDPR). Data are available from the Ethics Committee (contact via

## Abstract

Leukocyte viability (determined by e.g. propidium iodide [PI] staining) is automatically measured by hematology analyzers to check for delayed bench time. Incidental findings in fresh blood samples revealed the existence of leukocytes with decreased viability in critically ill surgical patients. Not much is known about these cells and their functional and/or clinical implications. Therefore, we investigated the incidence of decreased leukocyte viability, the implications for leukocyte functioning and its relation with clinical outcomes. An automated alarm was set in a routine hematology analyzer (Cell-Dyn Sapphire) for the presence of non-viable leukocytes characterized by increased fluorescence in the PI-channel (FL3:630 ±30nm). Patients with non-viable leukocytes were prospectively included and blood samples were drawn to investigate leukocyte viability in detail and to investigate leukocyte functioning (phagocytosis and responsiveness to a bacterial stimulus). Then, a retrospective analysis was conducted to investigate the incidence of fragile neutrophils in the circulation and clinical outcomes of surgical patients with fragile neutrophils hospitalized between 2013–2017. A high FL3 signal was either caused by 1) neutrophil autofluorescence which was considered false positive, or by 2) actual non-viable PI-positive neutrophils in the blood sample. These two causes could be distinguished using automatically generated data from the hematology analyzer. The non-viable (PI-positive) neutrophils proved to be viable (PI-negative) in non-lysed blood samples, and were therefore referred to as 'fragile neutrophils'. Overall leukocyte functioning was not impaired in patients with fragile neutrophils. Of the 11 872 retrospectively included surgical patients, 75 (0.63%) were identified to have fragile neutrophils during hospitalization. Of all patients with fragile neutrophils, 75.7% developed an infection, 70.3% were admitted to the ICU and 31.3% died during hospitalization. In conclusion, fragile neutrophils occur in the circulation of critically ill surgical patients. These cells can be automatically detected during routine blood analyses and are an indicator of critical illness.

info@metcutrecht.nl) for researchers who meet the criteria for access to confidential data.

**Funding:** The authors received no specific funding for this work.

**Competing interests:** The authors have declared that no competing interests exist.

# Background

Neutrophils are the most abundant type of leukocyte found in the peripheral blood [1] and contribute to the host first line of defense against invading micro-organisms. A shortage of functional neutrophils can lead to the development of severe bacterial infections and sepsis [2,3]. Increased neutrophil death might be either a cause or a sign of such an infectious state.

A classic method to show the presence of dying leukocytes is staining of nuclear DNA with propidium iodide (PI). This method visualizes diminished membrane integrity as PI is impenetrable in healthy intact cells. Hence, a high PI signal is associated with late apoptosis or necrosis of the cells [4,5]. The fraction of viable PI-negative leukocytes is assessed during every routine blood analysis by the Cell-Dyn Sapphire hematology analyzer (Abbott Diagnostics, Santa Clara, USA). This fraction is expressed as the white cell viability fraction (WVF) in the range 0–1 with 1 being 100% viable [6]. In fresh blood samples, the WVF generally ranges from 0.97–1.0 [7]. The WVF decreases with prolonged bench time and 0.95 is used as cut-off point for quality control of timely processing of blood samples [7].

However, preliminary findings showed intriguing results concerning a decreased WVF in fresh blood samples. These incidental findings showed PI-positive neutrophils in the blood of patients with severe septic shock with fatal outcome. Similar decreased WVF's were found in fresh blood samples of severely injured patients who had developed organ dysfunction (multiple organ dysfunction syndrome and/or acute respiratory distress syndrome) (S1 Fig). To our knowledge, non-viable neutrophils in the circulation have never been described before [8]. However, neutrophils are considered very sensitive cells that are easily damaged by *ex vivo* manipulation [9]. Hence, it was unknown whether a decreased WVF was really caused by the presence of non-viable neutrophils in the peripheral blood of these patients or by an artificial mechanism caused by ex vivo manipulation.

Therefore, the aim of this study was to reassess the presence of non-viable neutrophils in surgical patients with a decreased WVF. The presence of such non-viable neutrophils could influence overall neutrophil function and the susceptibility to infections. Therefore, the secondary aim of the study was to investigate neutrophil function and clinical outcomes of surgical patients with a decreased WVF.

# Results

## Prospective analysis

**Cause decreased white cell viability fraction.** In total, 18 surgical patients with a WVF ≤ 0.95 were prospectively included. Cell-Dyn Sapphire scatterplots of these patients either showed an elongated neutrophil population alongside the PI-axis (n = 9, Fig 1B) or a PI-positive neutrophil population separate from the PI-negative neutrophil population (n = 9, Fig 1C). Image stream analysis showed that the elongated neutrophil population was not caused by PI staining of nuclear or extracellular DNA, but caused by neutrophil autofluorescence, since intracellular fluorescence was observed in the absence of any fluorochromes including PI (Fig 1B). On the other hand, image stream analysis showed nuclear PI staining in neutrophils from samples that contained a PI-positive neutrophil population separate from the PI-negative neutrophil population, confirming the presence of truly non-viable neutrophils in these blood samples (Fig 1C). In these blood samples, both early apoptotic (7AAD-negative/AnnexinV-positive) and late apoptotic/necrotic neutrophils (Vivid-positive, PI-positive and 7AAD-positive) were found (results not shown). Therefore, a decreased white cell viability fraction is either caused by neutrophil autofluorescence (false positive) or caused by non-viable neutrophils (true positive).

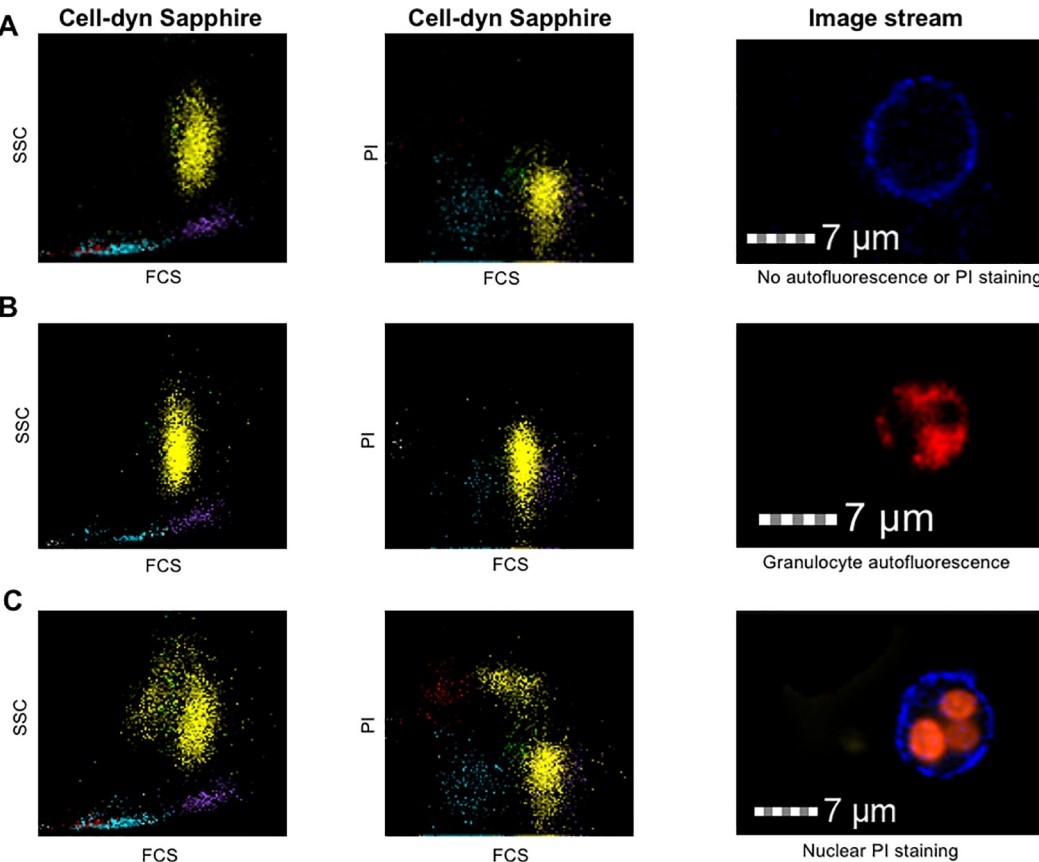

**Fig 1. Cell-dyn Sapphire light scatterplots and image stream figures.** (A) Cell-dyn Sapphire light scatterplots and image stream figures of a healthy control, (B) the same figures for a patient with autofluorescent neutrophils, and (C) the same figures for a patient with non-viable neutrophils in the blood sample. In patients with non-viable neutrophils, light scatter plots showed a PI-positive neutrophil population separate from the PI-negative neutrophil population and image stream analysis showed nuclear PI staining. In patients with autofluorescent neutrophils, light scatter plots showed an elongated neutrophil population alongside the PI-axis and image stream analysis showed neutrophil fluorescence without the addition of fluorochromes. Neutrophils in Fig 1A and Fig 1C were stained with both PI and surface marker CD16 prior to image stream analysis. Lymphocytes are light blue, monocytes are purple and granulocytes are yellow. PI = propidium iodide, FCS = forward scatter. SSC = side scatter.

**Non-viable neutrophils in vitro are fragile neutrophils in vivo.** When non-viable neutrophils were found during routine diagnostic blood sample analysis, these cells were only found after RBC lysis, both manually in the experimental blood samples (Fig 2C) as well as in blood samples processed by the Cell-Dyn Sapphire (Fig 2A). These non-viable neutrophils were not found during whole blood viability analysis (Fig 2B). This indicated that these patients had PI-negative neutrophils *in vivo*, that became PI-positive due to manipulation (i.e. RBC lysis) *in vitro*. Therefore, non-viable neutrophils *in vitro* are likely to be fragile, but viable neutrophils *in vivo*.

**Different neutrophil phenotype in patients with fragile neutrophils.** Patients with fragile neutrophils were found to have a lower neutrophil CD16 expression (without fMLF: $p = 0.046$, with fMLF: $p = 0.046$), a higher neutrophil CD64 expression (without fMLF: $p < 0.001$, with fMLF: $p < 0.001$) and a higher LAIR 1 expression (without fMLF: $p < 0.001$, with fMLF: $p = 0.001$) compared to healthy controls (Fig 3). Moreover, in patient samples with fMLF, CBRM1/5 was slightly higher ($p = 0.034$) and CD14 was slightly lower ($p = 0.023$) than in healthy controls.

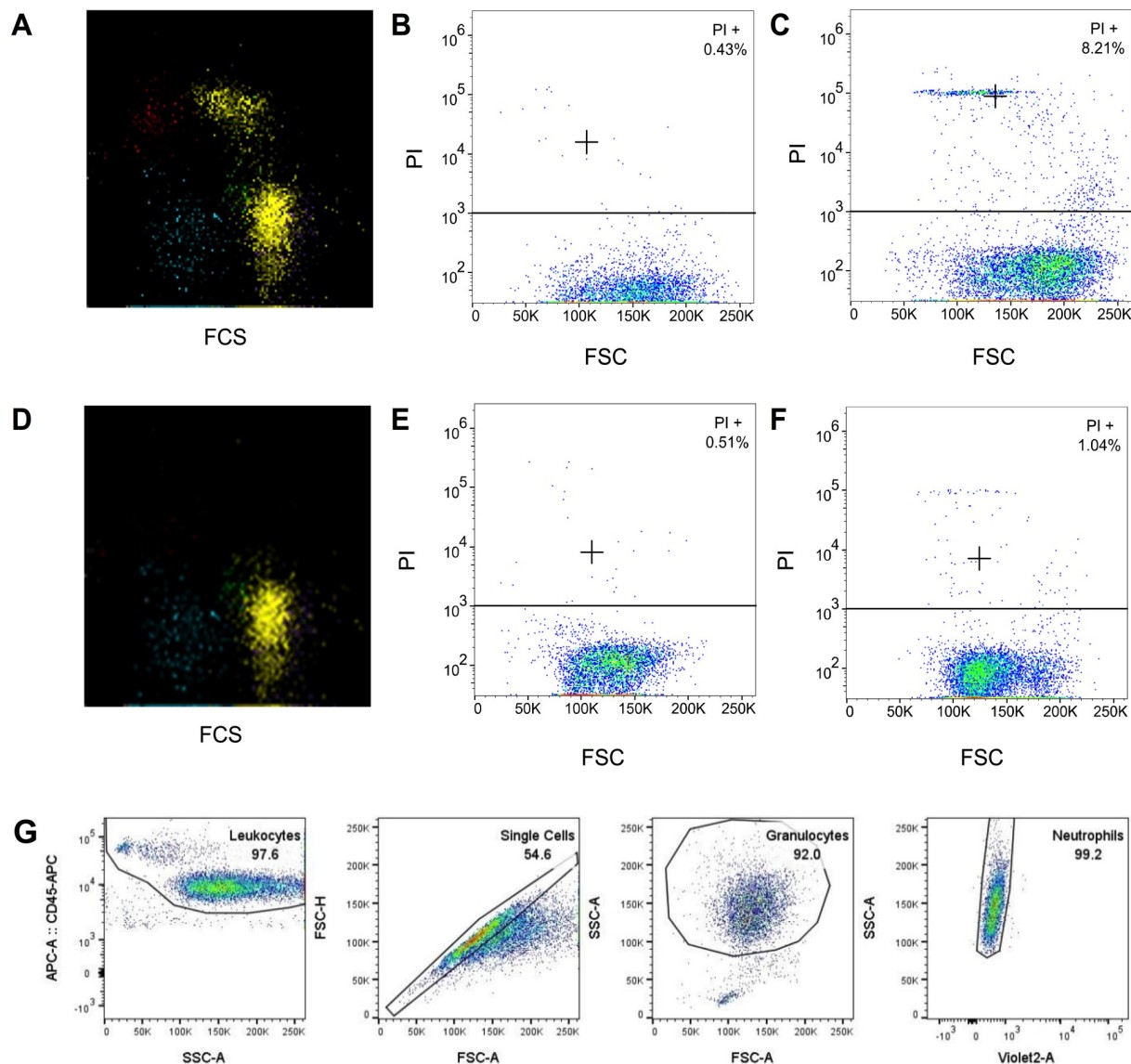

**Fig 2. Different viability outcomes depending on red blood cell lysis.** (A) Viability assay after automated PI staining and red blood cell lysis by the Cell-dyn Sapphire. (B) Viability assay after manual PI staining in whole blood. (C) Viability assay after manual PI staining and red blood cell lysis. (D-F) Same viability assays in a healthy control. (G) Gating strategy to identify neutrophils. Neutrophils are PI-negative in whole blood and a subset of neutrophils becomes PI-positive after manual lysis or lysis by the Cell-dyn Sapphire. Thus, non-viable neutrophils in vitro are fragile, but viable, neutrophils in vivo. PI = propidium iodide, FCS = forward scatter. SSC = side scatter.

**Overall neutrophil function was not impaired in patients with fragile neutrophils.** Fig 4A–4J shows neutrophil responsiveness to the bacterial stimulus fMLF in patients with fragile neutrophils and healthy controls. No significant differences were found between the two groups. Outcomes of the phagocytosis assays are depicted in Fig 4K and 4L. A significantly higher percentage of GFP-positive neutrophils was found in patients with fragile neutrophils compared to controls. This was found both with a MOI of 1 (20 minutes: $p = 0.022$, 40 minutes: $p = 0.035$) and with a MOI of 10 after 40 minutes ($p = 0.024$). The MFI of GFP-positive neutrophils did not differ between healthy controls and patients with fragile neutrophils.

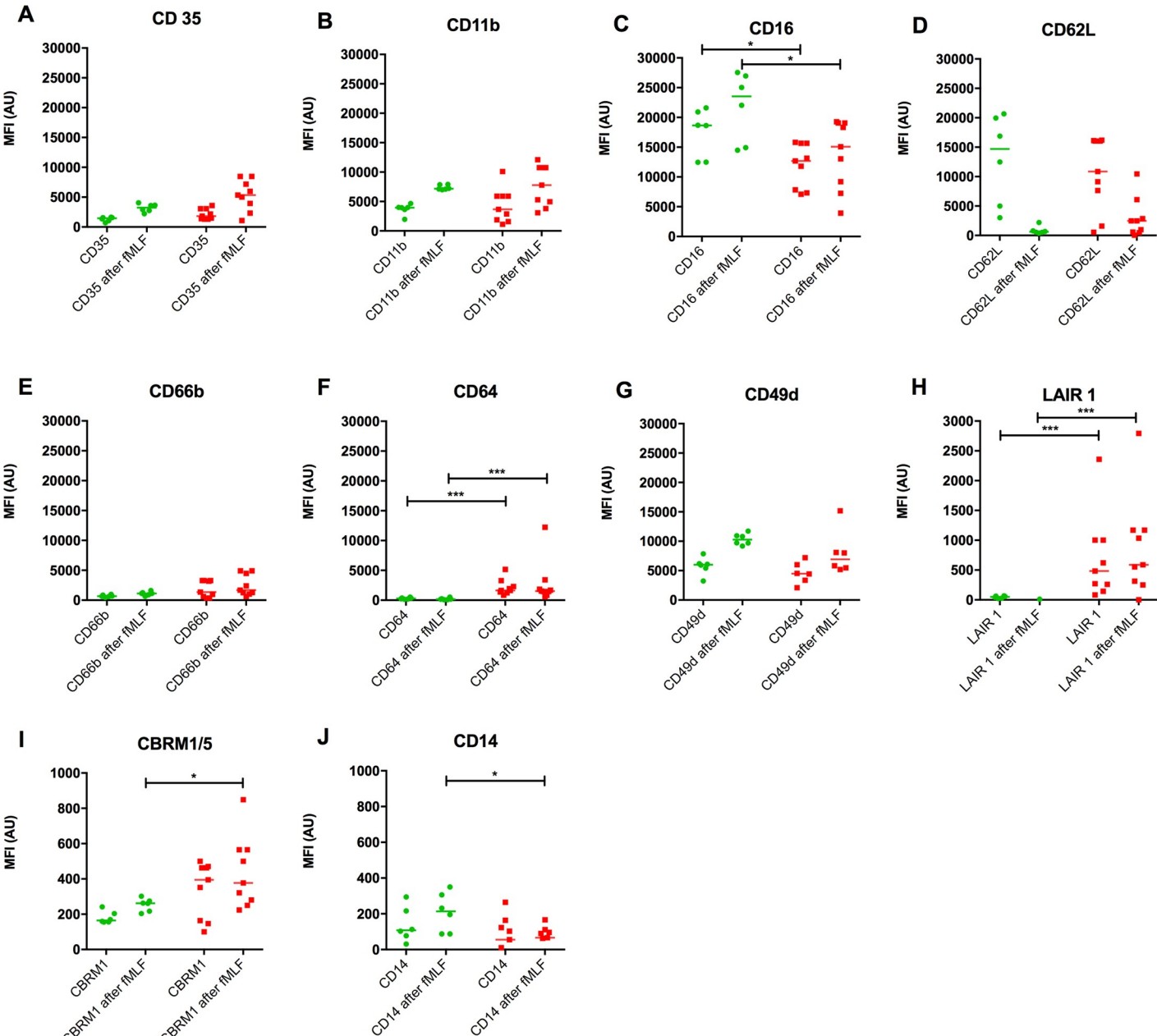

**Fig 3. Neutrophil receptor expression with and without bacterial stimulus.** (A-J) Neutrophil receptor expression with and without adding fMLF to blood samples of healthy controls (n = 6) (●) and patients with fragile neutrophils (n = 9) (■). Data are presented as scatter plot with median. Patients with fragile neutrophils were compared to healthy controls with the use of a Mann-Whitney $U$ test. MFI = median fluorescent intensity, fMLF = N-formyl-methionyl- phenylalanine, AU = arbitrary units. $^*P < 0.05$, $^{**}P < 0.005$, $^{***}P < 0.0005$.

### Retrospective analysis

**The presence of fragile neutrophils is rare in surgical patients.** In total, 13 760 surgical patients were eligible for inclusion. Of these patients, 43 were excluded because of clozapine use, 635 were excluded because of immunosuppressive drugs and 1 211 were excluded because of cytotoxic drugs. This led to a total number of 11 871 patients who were further analyzed. Of these patients, there were 257 patients (2.2%) with a decreased WVF during hospitalization.

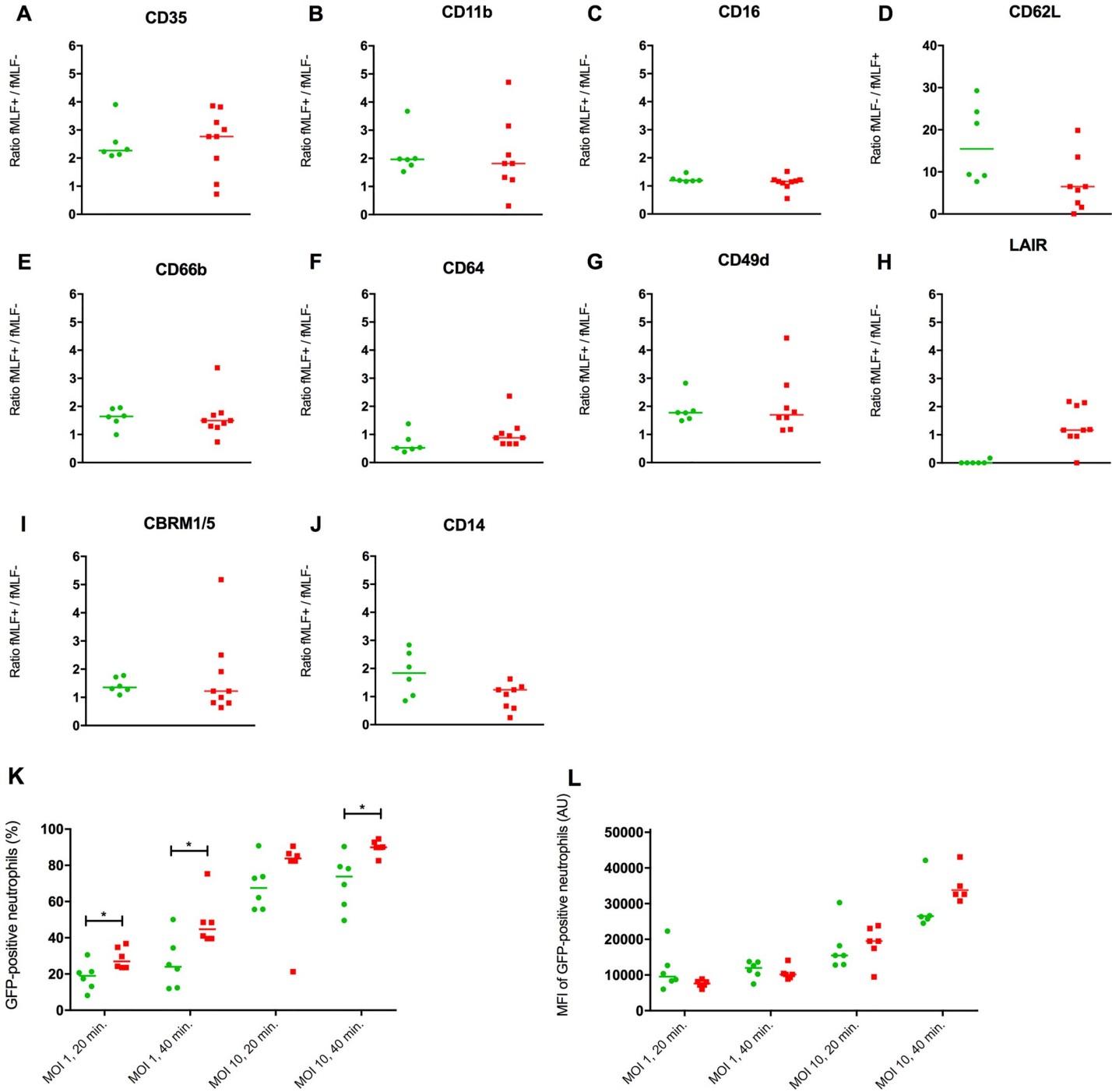

**Fig 4. Neutrophil phagocytosis and responsiveness to a bacterial stimulus.** (A-J) Neutrophil responsiveness to fMLF in healthy controls (n = 6) and patients with fragile neutrophils (n = 9). Responsiveness of neutrophils was defined as (MFI [AU] sample with fMLF) / (MFI [AU] sample without fMLF) for all receptors except for CD62L. CD62L is shed after stimulation with fMLF, and responsiveness was therefore defined as (MFI sample without fMLF) / (MFI sample with fMLF). (K) Percentage of GFP-positive neutrophils in healthy controls (n = 6) and patients with fragile neutrophils (n = 6) after incubation with *S. Aureus*-GFP. (L) GFP MFI of GFP-positive neutrophils after incubation with *S. Aureus*-GFP. Healthy controls are depicted in green (●) and patients with fragile neutrophils in red (■). Data are presented as scatter plot with median. Patients with fragile neutrophils were compared to healthy controls with the use of a Mann Whitney U test. MFI = median fluorescent intensity, fMLF = N-formyl-methionyl-phenylalanine. *S. Aureus* = *Staphylococcus Aureus*, GFP = Green fluorescent protein, AU = arbitrary units, MOI = multiplicity of infection. *P < 0.05.

The causes for this decreased WVF were autofluorescent neutrophils in 127 patients (49%), fragile neutrophils in 84 patients (33%), an extended lysing procedure in 17 patients (6.6%) and cellular debris that could not be classified in 29 patients (11%). After checking the electronic health records of the 84 patients with fragile neutrophils, 5 were excluded because the decreased WVF was associated with a delay in bench time and 5 were excluded because the decreased WVF was associated with a sample error. Hence, a total of 74 patients of the 11 871 eligible patients (0.62%) were identified to have truly fragile neutrophils during hospitalization.

**Fragile neutrophils are associated with critical illness.** Surgical patients with fragile neutrophils were somewhat older (median of 65.0 versus 62.0 years, p = 0.011). Also, patients with fragile neutrophils more often received corticosteroids during admission (7.6% versus 14.9%, p = 0.027). However, only two (2.7%) patients received corticosteroids at the moment the alarm went off and no time-dependence was observed between the start of corticosteroids and the day the alarm went off. Patients with fragile neutrophils had a longer hospital stay (median of 30 versus 3.0 days, p < 0.001), a longer ICU stay (median of 6.3 versus 0.0 days, p < 0.001), a higher ICU admission rate (70.3% versus 7.9%, p < 0.001) and a higher mortality rate (31.1% versus 1.5%, p < 0.001) than other surgical patients (Table 1). Causes of death were abdominal

**Table 1. Baseline characteristics and clinical outcomes of surgical patients with fragile neutrophils.**

| | Patients with normal leukocyte viability (n = 11,610) | Patients with ragile neutrophils (n = 74) | P-value |
|---|---|---|---|
| Gender, male | 6571 (56.6%) | 47 (63.5%) | 0.242 |
| Age | 62.0 (48.0–72.0) | 65.0 (54.0–77.0) | **0.017** |
| Hospital specialism | | | 0.067 |
| trauma surgery | 2897 (25.0%) | 24 (32.4%) | |
| vascular surgery | 3467 (29.9%) | 13 (17.6%) | |
| gastrointestinal/oncologic surgery | 3137 (27.0%) | 25 (33.8%) | |
| general surgery | 2109 (18.2%) | 12 (16.2%) | |
| Hospital length of stay | 3.0 (1.0–8.0) | 30 (11.0–53.0) | < **0.001** |
| ICU admission | 914 (7.9%) | 52 (70.3%) | < **0.001** |
| ICU length of stay (days) | 0.0 (0.0–0.0) | 6.3 (0–14.4) | < **0.001** |
| Hospital mortality | 176 (1.5%) | 23 (31.1%) | < **0.001** |
| Percentage of non-viable leukocytes | | 7.0 (6.0–10.3) | |
| Cause of death | | | |
| Abdominal sepsis | | 10 (13.3%) | |
| Pneumonia | | 4 (5.3%) | |
| Line sepsis | | 2 (2.7%) | |
| Sepsis of unknown origin | | 5 (6.7%) | |
| Bleeding | | 1 (1.3%) | |
| Intracerebral hemorrhage | | 1 (1.3%) | |
| Patients with complications **during admission** | | 61 (82.4%) | |
| Complications observed: | | | |
| Infection | | 56 (75.7%) | |
| Acute kidney injury | | 10 (13.5%) | |
| Cardiac arrest | | 4 (5.4%) | |
| Rhabdomyolysis | | 3 (4.1%) | |
| Severe bleeding | | 2 (2.7%) | |
| Pulmonary embolism | | 2 (2.7%) | |
| Acute respiratory distress syndrome | | 2 (2.7%) | |
| Multiple organ dysfunction syndrome | | 1 (1.4%) | |
| Acute limb ischemia | | 1 (1.4%) | |

(*Continued*)

**Table 1.** (Continued)

| | Patients with normal leukocyte viability (n = 11,610) | Patients with ragile neutrophils (n = 74) | P-value |
|---|---|---|---|
| Patients with complications **during viability alarm** | | 53 (73.0%) | |
| Complications observed: | | | |
| Infection | | 45 (60.8%) | |
| Acute kidney injury | | 9 (12.2%) | |
| Cardiac arrest | | 3 (4.1%) | |
| Severe bleeding | | 2 (2.7%) | |
| Rhabdomyolysis | | 2 (2.7%) | |
| Pulmonary embolism | | 2 (2.7%) | |
| Acute respiratory distress syndrome | | 2 (2.7%) | |
| Multiple organ dysfunction syndrome | | 1 (1.4%) | |
| Acute limb ischemia | | 1 (1.4%) | |

Data are presented as median with interquartile range (IQR) or n (%). Variables are compared between patients with normal leukocyte viability and patients with fragile neutrophils with a Fisher's exact test or a Mann-Whitney U test, as indicated. ICU = intensive care unit.

sepsis (n = 10) pneumonia (n = 4), line sepsis (n = 2), sepsis of unknown origin (n = 5), severe bleeding (n = 1) and intracerebral hemorrhage (n = 1). Of all patients with fragile neutrophils, 61 (82.4%) experienced complications during their admission, of whom 56 developed an infectious complication. In 53 patients with fragile neutrophils (73.0%), a decreased WVF was found during a complication, of which 85% (n = 45) was an infectious complication. Of the 21 patients who did not have a diagnosed complication during the decreased WVF, 12 suffered from multiple injuries due to high energetic trauma, 4 had surgeries on the day the alarm went off, 1 developed a pneumonia 3 days later, in 1 patient myocardial ischemia was suspected and 1 patient experienced unexplained loss of consciousness several times that day.

## Discussion

A decreased WVF, as automatically measured by hematology analyzers such as the Cell-Dyn Sapphire, is caused by either neutrophil autofluorescence (false positive) or by fragile circulating neutrophils (true positive). Fragile neutrophils were found to be cells that were PI-negative *in vivo* and became PI-positive only after minimal manipulation *in vitro*.

To our knowledge, fragile neutrophils detected by automated hematology analysis, have never been described in the literature before. It has been published that leukocytes are more sensitive to *in vitro* manipulation in certain clinical conditions, such as chronic lymphocytic leukemia [10]. For CLL it is well known that damaged lymphocytes and neutrophils, or so-called "smudge cells", can be observed in peripheral blood smears. A recent study described that these cells can also be present in peripheral blood smears of patients with other disorders, including cardiac arrest, infections, solid cancers and other hematological malignancies [11]. However, the preparation of blood smears is a manual process that is sensitive to *ex vivo* manipulation, and is, therefore, difficult to relate to our findings of non-viable neutrophils found after red blood cell lysis. There are a few studies that did describe the occurrence of non-viable neutrophils after red blood cell lysis. These studies focused on non-viable neutrophils in patients with trauma or severe inflammation [12–14]. However, these studies were based on *in vitro* findings after manual work-up of blood samples and the high numbers of apoptotic and necrotic neutrophils (up to 99%) found in these studies suggest cell death due to *in vitro* manipulation. In general, it is difficult to interpret studies on fragile neutrophils

because small changes in the work-up and analysis of blood samples can easily affect the presence and number of these cells. Our study circumvented these operational problems by applying a fast, standardized and fully automated hematology analyzer to test leukocyte viability.

By using this standardized analysis method, we found that the concept of fragile neutrophils relates to a true phenomenon. However, it is very rare (< 1% of all surgical patients) and these cells are only found in the most severely ill surgical patients. Most of these patients were admitted to the ICU and one-third of these patients died during hospital admission. Fragile neutrophils were mostly detected in surgical patients with recurrent or serious infections. On the other, fragile neutrophils were also observed without infection in patients who sustained high energy trauma and in patients with multiple or major surgeries. It is tempting to speculate that these conditions all cause significant systemic inflammation with a large consumption of neutrophils [15]. Neutrophil lifespan is 4–5 days and it takes 7 days for neutrophils to mature in the bone marrow and migrate into the blood [16]. Therefore, when large quantities of neutrophils are released into the blood in response to (systemic) inflammation, this could hypothetically lead to a decrease in viable neutrophils between day 5 and 7. This concept is supported by the finding that fragile neutrophils were mostly found after several days of inflammation or after a "second hit" (a second inflammatory stimulus such as major surgery after trauma [17]). An additional supportive finding was that neutrophil counts decreased before the number of non-viable leukocytes increased (S2 Fig). Although neutrophil numbers remained high-normal during this decrease, it is possible that the high consumption rate led to a relative neutrophil shortage for patients in such an inflammatory state, possibly initiating an unknown signal to keep less viable neutrophils in the circulation. Further research should investigate whether these fragile neutrophils are indeed an indication of immune exhaustion due to neutrophil (over)consumption.

Microscopic blood smear analysis of patients with fragile neutrophils revealed the presence of many banded neutrophils, neutrophil progenitors, and sometimes cells with toxic granulation and neutrophils with cytoplasmic vacuoles. (example in S3 Fig) These characteristics are typical for severe inflammation and/or infection [18,19] and can sometimes be seen after administration of corticosteroids [20]. However, the prospectively included patients who displayed these characteristics did not receive corticosteroids. Furthermore, neutrophil receptor expression supported the hypothesis of the presence of severe inflammatory processes in these patients. A decreased CD16 was found in combination with an increase in LAIR 1, which suggested the release of immature neutrophils from the bone marrow in these patients [16,21,22]. Furthermore, CD64, an indicator of infection [23,24], was higher in most patients compared to healthy controls. This corresponded with our finding that a significant portion of patients with fragile neutrophils experienced infectious complications when the viability alarm went off. Also, the activation marker CBRM1/5 (binds to active CD11b) was higher in patients [25,26] and membrane bound CD14 was slightly lower in patients, which was suggestive of inflammation and neutrophil activation [27,28]. No correlation was found between the WVF and neutrophil receptor expression (S1 Table). Altogether these findings suggest that the phenotypical neutrophil changes observed in patients with fragile neutrophils are an indicator of severe systemic inflammation, rather than specific characteristics of fragile neutrophils.

These preliminary data on neutrophil functionality (responsiveness to fMLF and phagocytosis of bacteria) suggest that neutrophil function was not impaired in the prospectively included patients with a decreased WVF. On the contrary, patients showed enhanced phagocytosis compared to healthy controls. Conflicting results on this subject have been reported in literature. Inflammation has been associated with a decrease in neutrophil phagocytosis [29,30], no significant difference in phagocytosis [31,32] and increased phagocytosis [33]. A possible explanation for these differences was postulated by Taneja *et al.* [31], who found that banded

neutrophils had impaired phagocytic capacity and these authors thus speculated that the number of neutrophil subtypes influenced the outcome of the phagocytosis assay. This hypothesis did not correspond to our findings since we found a lower neutrophil CD16 expression in patients compared to healthy control that is associated with a higher number of banded neutrophils [34]. Since the phagocytosis assay was performed in undiluted whole blood and most patients had a leukocytosis during blood sampling (S2 Fig; blood sampling between day 0 and 1), we hypothesized that the increase in GFP-positive neutrophils was not due improved neutrophil function, but rather due to an increase in neutrophil count. The MOI was kept constant in the experiments, so an increase in neutrophil count led to an increase in bacteria count. It is likely that higher neutrophil and bacteria numbers in a shaking suspension of constant volume led to a higher percentage of GFP-positive neutrophils simply because the chance of neutrophils encountering bacteria increased. This hypothesis was supported by our data showing a decrease in the percentage of GFP-positive neutrophils after dilution of whole blood with plasma from the same sample (S4 Fig).

Although overall neutrophil function was not impaired in patients with fragile neutrophils, it remains unknown how the actual fragile neutrophils function. Functional characterization of such fragile neutrophils was not possible as they could not be sorted by FACS sorting, because we did not succeed to identify these cells by a specific receptor expression profile. We could only identify fragile neutrophils by viability stains. Unfortunately, once the cells had become positive for these stains, sorting was trivial as membrane integrity of these cells was diminished. A limitation of this study was the small prospective study group, because the presence of fragile neutrophils is so very rare and because this was a proof-of-principle study to describe the phenomenon. Due to this, data on neutrophil functionality and neutrophil receptor expression in these patients is still preliminary. Also no other inflammatory mediators, such as cytokines, were studied. These mediators could help to gain better insight into the extent of systemic inflammation in patients with fragile neutrophils. Future studies should investigate functional neutrophil tests in combination with other inflammatory mediators in a larger cohort to gain better insight in how the presence of fragile neutrophils influences overall neutrophil functionality. Another limitation was that several patients could not be prospectively included because no informed consent was obtained. These were often the patients who were most critically ill. To circumvent the possibility of a selection bias we chose to separately analyze the incidence of fragile neutrophils and clinical outcomes associated with this phenomenon in a retrospective cohort consisting of all surgical patients of the last 5 years. To our knowledge, no clinical biomarker exists that coincides with both the presence of fragile cells and severe critical illness such as found in this study. Although further research into the pathophysiology behind fragile neutrophils is needed, this study clearly points out the association between these cells and clinical deterioration. Moreover, this biomarker is particularly suitable for clinical practice, since the WVF is automatically analyzed during every blood analysis irrespective of the requested parameter. Blood samples are frequently drawn from critically ill patients and, therefore, the assessment of fragile neutrophils would not require any additional blood samples.

To facilitate implementation of this analysis into routine blood analysis, we sought for an algorithm to automatically distinguish samples with actual non-viable neutrophils from samples with autofluorescent neutrophils based on automatically generated data from the Cell-Dyn Sapphire. When the whole neutrophil population is elongated alongside the PI-axis (autofluorescent neutrophils), both the mean and the median PI fluorescence significantly increase. However, in samples with a separate PI-positive neutrophil population (fragile neutrophils), the mean PI fluorescence significantly increases, whereas the median is less affected by the skewness of the population. Therefore, by dividing the mean PI fluorescence by the median PI

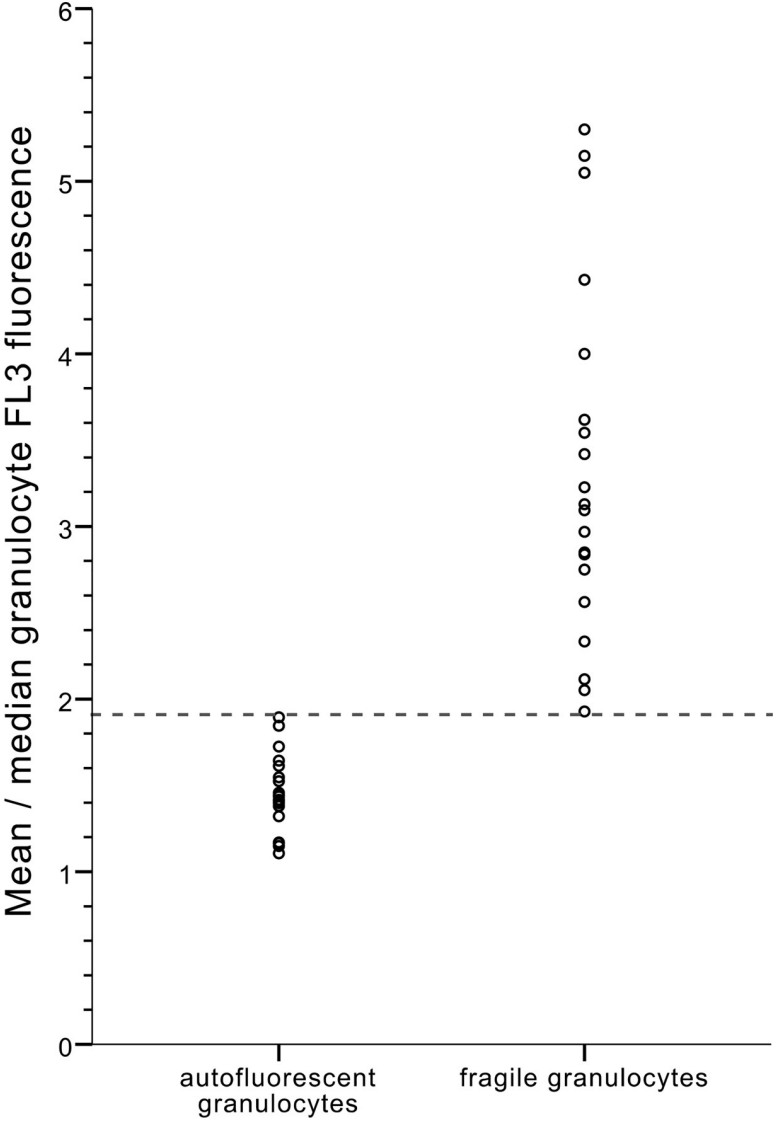

**Fig 5. Ratio to discriminate samples with autofluorescent granulocytes from samples with fragile granulocytes.**
Performance of ratio to discriminate between samples with autofluorescent granulocytes (n = 20) and samples with fragile granulocytes (n = 20) based on automatically generated Cell-Dyn Sapphire data. By dividing mean granulocyte PI fluorescence by median granulocyte PI fluorescence, a perfect discrimination between samples with autofluorescent granulocytes (ratio < 1.9) and samples with fragile granulocytes (ratio > 1.9), can be reached. A receiver operating characteristic curve was generated with an area under the curve of 1.00 (95%CI: 1.00–1.00). FL3 is the Cell-Dyn Sapphire's third fluorescent channel (630±30 nm) in which propidium iodide is measured.

fluorescence, a perfect discrimination between samples with autofluorescent neutrophils (ratio < 1.9) and samples with fragile neutrophils (ratio > 1.9), can be reached (Fig 5). Both the median and the mean neutrophil PI fluorescence are parameters that can be automatically generated by the Cell-Dyn Sapphire. Therefore, by implementing this simple ratio (mean PI fluorescence/ median PI fluorescence) into the hematology analyzer, it is probably possible to automatically detect fragile neutrophils in every blood sample measured within the hospital. Although validation of this ratio in a more extensive prospective cohort is needed, this study suggests that the ratio could provide valuable additional information about the clinical condition of a severely ill patient.

In conclusion, fragile neutrophils circulate in the peripheral blood of surgical patients and can be automatically detected during every routine blood analysis. The origin and function of these cells remain to be further elucidated. Although it is unknown if there is a causal link between fragile neutrophils and adverse clinical outcomes, it is clear that the presence of these cells holds valuable information about the clinical condition of surgical patients. The presence of these neutrophils is very rare, but if these cells are detected, this is associated with critical illness and severe clinical deterioration.

## Materials and methods

### Study design

A prospective analysis was carried out in which surgical patients with a WVF $\leq$ 0.95 were included to reassess leukocyte viability and leukocyte functioning. Based on this analysis, we were able to determine which patients had a true positive decreased WVF due to actual non-viable neutrophils in the sample and which measurements were false positive (due to auto-fluorescence). Secondly, a retrospective analysis was performed to investigate the incidence of a true positive decreased WVF and investigate its consequences for clinical outcomes in surgical patients.

### Prospective analysis

**Patients.** Surgical patients ($\geq$ 18 years of age) with a WVF $\leq$ 0.95 in routine diagnostic blood samples were prospectively included between October 2017 and April 2019 in the University Medical Center (UMC) Utrecht, the Netherlands. These patients were recruited from the departments of general surgery, trauma surgery, vascular surgery, surgical oncology, the surgical intermediate care unit and intensive care unit (ICU) KG. Exclusion criteria were present or recent ($<$ 3 months) use of immunosuppressive or cytotoxic medication, a known HIV-positive status or other immunosuppressive diseases, and the use of clozapine, since this is associated with an increased FL3 signal [35]. Control blood samples were provided by anonymous, sex and age matched healthy volunteers. Blood samples were only collected after written informed consent of the subject or legal representative in accordance to the Declaration of Helsinki. In addition, when proxy consent was obtained and the patient recovered to good mental health, the patient also provided informed consent for the use of the previously collected blood samples for research purposes. All experiments were performed in accordance with the relevant guidelines and regulations. This study was approved by the UMC Utrecht ethical review committee (17–546). The trial was registered by the Central Committee on Research Involving Human Subjects in The Netherlands before participant enrollment started (NL60543.041.17).

**Viability alarm and experimental blood sampling.** Diagnostic blood samples from surgical patients were drawn as indicated by their treating physician. Samples were routinely analyzed on the Cell-Dyn Sapphire hematology analyzer. This is an automated hematology analyzer that uses spectrophotometry, electrical impedance, laser light scattering and 3 color fluorescent technologies to classify blood cells [36]. The Cell-Dyn Sapphire is a top of the range hematology analyzer that is found to have an excellent precision, linearity and inter-instrument agreement [37,38]. During every blood analysis PI is added for determination of the WVF in the third fluorescent channel (FL3: 630±30 nm). For this study, an alarm was set on samples with a WVF $\leq$ 0.95 ($\leq$ 95% viable leukocytes) after which a notification was sent to the research team. When bench time of the diagnostic blood sample was not prolonged (measured $<$ 60 minutes) and the patient met the inclusion criteria, he/she was enrolled after written informed consent was obtained. Since leukocyte characteristics change rapidly, new

blood samples were drawn as soon as possible but no later than 3.5 hours after the alarm. Two 4 milliliter sodium heparin blood tubes were obtained for viability staining and functional assays. To check if the decreased white cell viability fraction (WVF) was not a results of processing time and to check if the fragile cells were still present in the peripheral blood of the patient, this new blood sample was again tested for the presence of fragile neutrophils by the hematology analyzer. This second test was always performed within 15 minutes, so that there was basically no time delay between the blood drawing and the start of the sample work-up.

**Data obtained from the hematology analyzer Cell-Dyn Sapphire.** Cell-Dyn Sapphire light scatterplots and FL3 plots of included patients were obtained to analyze the characteristics of leukocytes with a high FL3 signal. These light scatter plots were automatically generated during routine diagnostic blood sampling. The FL3 signal was analyzed to confirm or negate the existence of a PI-positive neutrophil population separate from the PI-negative neutrophil population, indicating the presence of actual non-viable neutrophils in the blood sample (true positive) and excluding other causes for a high FL3 signal such as autofluorescence (false positive). Also, scatter characteristics were analyzed to investigate whether an algorithm could be developed to automatically distinguish true positive from false positive samples.

**Viability assays.** Leukocyte viability was reassessed in the experimental blood sample (sodium heparin tube) both in whole blood and after lysis of red blood cells (RBCs) with ammonium chloride ($NH_4Cl$) hemolysis solution [39]. For the whole blood viability analysis, whole blood was stained with PI (Sigma-Aldrich, St. Louis, USA) and with CD45-APC (Clone 2D1, APC-labeled; BD Biosciences, San Jose, USA) for recognition of leukocytes during flow cytometry analysis. For the viability analysis after hemolysis, RBCs were lysed in isotonic ice-cold $NH_4Cl$ solution. Then, leukocytes were washed twice and resuspended in Dulbecco's phosphate buffered saline (Merck KgaA, Darmstadt, Germany) complemented with pasteurized plasma solution (10%) and trisodium citrate (0.4%[wt/vol]) (PBS2+) to a concentration of $10 \times 10^6$ cells/ml for flow cytometric analysis and to $20 \times 10^6$ cells/ml for ImageStream analysis (Amnis® ImageStream®XMk II, Luminex, Austin, USA). Then, leukocytes were stained with PI (Sigma-Aldrich, St. Louis, USA), Vivid LIVE/DEAD® fixable violet dead cell stain (Thermo Fisher Scientific, Waltham, USA), AnnexinV (BD Biosciences) and 7-aminoactinomycin D (7AAD) (BD Biosciences) for flow cytometry analysis. During early apoptosis phosphatidylserine is transferred from the inner leaflet of the plasma membrane to the outer leaflet and becomes recognized by AnnexinV. During late apoptosis and necrosis the membrane integrity is compromised and leukocytes become permeable for small molecules [40]. As a consequence cell become positive for Vivid (staining free amines in the cytoplasm) and for 7AAD and PI (staining DNA) [4,40–43]. Double staining with AnnexinV/7AAD was used to assess early apoptosis and late apoptosis/necrosis in the same sample. Additionally, vivid and PI were used to assess late apoptosis and necrosis. Moreover, leukocytes were stained with PI for ImageStream analysis to confirm actual nuclear PI staining and exclude other causes for PI-positive leukocytes, such as staining of extracellular free DNA [44–46]. For this analysis, samples were run at 100 cells per second on the Amnis® ImageStream®XMk II and the data were analyzed using the ImageStream Data Analysis and Exploration Software (IDEAS, Luminex, Austin, USA).

**Neutrophil phagocytosis.** The phagocytosis assay was performed in whole blood. First, leukocyte count and differentiation were measured using the Cell-Dyn automatic hematology analyzer. Then *Staphylococcus Aureus* strain expressing green fluorescent protein (*S. Aureus*-GFP) [47] was added to samples with a bacteria-to-phagocytes ratio (multiplicity of infection [MOI]) of 1 and 10 based on measured phagocyte numbers. The whole blood with bacteria was shaken for 40 minutes at 37<C. After 20 and 40 minutes, part of the sample was removed and put on ice. Leukocytes were stained for 15 minutes on ice with CD45-APC (Clone 2D1,

APC-labeled; BD Biosciences) for identification during analysis. After staining, cells were fixed with paraformaldehyde 1% (wt/vol). The phagocytic capacity of neutrophils was analyzed by determining the percentage of neutrophils containing *S. Aureus*-GFP (GFP-positive neutrophils) and by measuring the median fluorescence intensity (MFI) of GFP-positive neutrophils.

**Neutrophil phenotype and responsiveness.** This assay was performed on total leukocytes after RBC lysis, as described above. Leukocytes were resuspended in PBS2+ to a concentration of $10 \times 10^6$ cells/ml and divided over two samples. One sample was incubated with 1μM N-for-myl-methionyl-leucyl-phenylalanine (fMLF) for 15 minutes in a 37<C water bath and the other sample was placed in the same water bath without the addition of fMLF. After incubation, cells were stained with antibody-fluorochrome conjugates for the markers CD16 (clone 3G8, Krome Orange labeled; Beckman Coulter, Fullerton, USA), CD62L (clone DREG-56, Brilliant Violet 650 labeled; Sony Biotechnology, San Jose, California), CD11b (clone BEAR1, Alexa Fluor 750 labeled; Beckman Coulter), CD35 (clone E11, FITC labeled; BD Bioscience), CD64 (clone 10.1, Alexa Fluor 647 labeled; Sony Biotechnology), CD66b (clone G10F5, PerCP-Cy5.5 labeled; Sony Biotechnology), CD49D (clone 9F10, PE/Cy7 labeled, Biolegend, San Diego, USA), LAIR-1 (clone DX26, PE labeled; BD Bioscience), active CD11b (clone CBRM1/5, Alexa Fluor 700 labeled; eBioscience, ThermoFisher, Waltham, USA) and CD14 (clone 61D3, eFluor 450 labeled; Invitrogen, ThermoFisher). Leukocytes were kept on ice in the dark for 30 minutes, after which cells were washed once and resuspended in 100μL para-formaldehyde 1% (wt/vol). The expression of neutrophil markers was quantified and expressed as MFI. Responsiveness of all receptors, except CD62L, was defined as (MFI sample with fMLF) / (MFI sample without fMLF). Responsiveness of CD62L, which is shed after stimulation with fMLF, was defined as (MFI sample without fMLF) / (MFI sample with fMLF).

**Flow cytometry analysis.** Samples from the viability assay were measured on the BD FACSCanto™ II (BD Biosciences). Samples from the phagocytosis assay and the fMLF responsiveness assay were measured on the BD LSRFortessa™ (BD Biosciences). For all experiments a minimum of 10 000 leukocytes were measured. In case of whole blood, leukocytes were identified based on their CD45 expression. In all samples, granulocytes, monocytes, and lymphocytes were identified based on their specific forward/side scatter patterns. Neutrophils were identified by excluding granulocytes that demonstrated eosinophil-specific autofluorescence or low CD16 expression. Flow cytometry data was analyzed with FlowJo® v10 software (FlowJo, LLC, Ashland, USA).

## Retrospective analysis

**Patient selection.** A retrospective analysis was conducted to analyze the incidence of a decreased WVF in surgical patients and the relation with clinical outcomes of these patients. All surgical patients ≥ 18 years of age admitted to the UMC Utrecht between January 1, 2013, and December 31, 2017, were included. Again, patients were excluded if they used clozapine, cytotoxic or immunosuppressive medication. Surgical patients were selected from the Utrecht Patient Oriented Database (UPOD). The technical details of the UPOD have been described previously [48]. In short, this database is an infrastructure of relational databases that allows (semi)automated transfer, processing and storage of data, including administrative information, medical and surgical procedures, medication orders, and laboratory test results for all clinically admitted patients and patients attending the outpatient clinic of the UMC Utrecht since 2004. Laboratory test results came from the Cell-Dyn Sapphire hematology analyzer [48]. The reliability and validity of laboratory results were monitored through routine quality control. The process and storage of data were in accordance with privacy and ethics regulations. Because no extra material, such as blood samples, was taken from patients and the data were

completely anonymized, there was no requirement to obtain informed consent from individual patients. A waiver was provided by the institutional medical ethics committee for this retrospective analysis. In addition, in line with the academic hospital policy, an opt-out procedure was in place for use of patient data for these research purposes.

**Determination of the incidence and causes of a decreased WVF after surgery.** Original flow cytometric data files of the Cell-Dyn Sapphire hematology analyzer were obtained. All data files of patients with a decreased WVF were analyzed to investigate the cause of the decreased WVF. Furthermore, anonymized electronic health records of patients with a decreased WVF, that was not repeatedly decreased, were checked for other causes such as a delay in bench time and sample errors. For example, a sample error was recorded if the sample with the decreased WVF was not recorded in the electronic health record and instead another blood sample was obtained that did not have a decreased WVF. Also, it was investigated if there was a relation between administration of corticosteroids and a decreased WVF, since corticosteroids are known to interfere with neutrophil apoptosis [49,50]. Afterwards the incidence of fragile neutrophils was determined for different surgical patients.

**Clinical outcomes.** Baseline characteristics (gender, age, surgical specialty) and clinical outcomes of patients with fragile neutrophils were recorded. These outcomes included ICU admission, mortality and length of stay in hospital and/or ICU. Additionally, significant complications (Clavien-Dindo $\geq$ II) [51] and cause of death were recorded for patients with a decreased WVF.

## Statistical analysis

Data were analyzed with IBM SPSS version 23 (IBM Corporation, NY, United States) and Graphpad Prism version 5 (GraphPad, La Jolla, United States). The distribution of continuous variables was assessed with the use of the Kolmogorov-Smirnov test. Clinical outcomes and demographics were compared between outcome groups using a Fisher's exact test or a Mann-Whitney $U$ test, as indicated. Neutrophil phagocytosis and neutrophil responsiveness to fMLF were compared between patients and healthy control donors. For these analyses a Mann-Whitney $U$ test was used, because the data were not normally distributed. A receiver operating characteristic curve with an area under the curve was generated to analyze the predictive value of the algorithm that distinguishes samples with non-viable neutrophils (true positive decreased WVF) from samples with autofluorescent neutrophils (false positive decreased WVF). Statistical significance was defined as a p-value < 0.05.

## Supporting information

**S1 Fig. White cell viability fraction in polytrauma patients developing organ dysfunction.** (A) White cell viability fraction in polytrauma patients over days after trauma. (B) White cell viability fraction in polytrauma patients who develop organ dysfunction relative to the first day organ dysfunction became clinically evident. Patients with organ dysfunction (n = 11) are depicted in red (▲) and patients without organ dysfunction (n = 70) are depicted in green (●). Organ dysfunction is defined as acute respiratory distress syndrome and/or multiple organ dysfunction syndrome. In Fig 1B, day 0 is the first day that organ dysfunction became clinically evident. Data are presented as mean with standard error of the mean (SEM).
(PDF)

**S2 Fig. Total leukocyte count and non-viable leukocyte count over time in patients with fragile neutrophils.** Leukocyte count (●) and non-viable leukocyte count (▲) over time in patients with fragile neutrophils (n = 9). Day 0 represents the first day that the white cell

viability fraction was ≤ 0.95 (dotted line). Leukocyte numbers decreased before the number of non-viable leukocytes increased. Still, leukocyte counts were above reference values (reference range adults: 4.0–11 x 109 / L) for almost the entire period. Data are presented as mean with standard error of the mean.
(PDF)

**S3 Fig. Examples of leukocyte cytospins of patients with fragile neutrophils.** Leukocyte cytospins of patients with fragile neutrophils revealed remarkable morphological neutrophil characteristics that are associated with severe inflammation. Firstly, banded neutrophils and other progenitors were often observed in these cytospins. Secondly, some cytospins contained neutrophils with cytoplasmic vacuoles and toxic granulation.
(PDF)

**S4 Fig. Percentage of GFP-positive neutrophils in different dilutions.** (A) Percentage of GFP-positive neutrophils in different dilutions measured after 20 minutes of incubation with *S. Aureus*-GFP. (B) Percentage of GFP-positive neutrophils in different dilutions measured after 40 minutes of incubation with *S. Aureus*-GFP. Whole blood samples of healthy controls (n = 5) were diluted with plasma obtained from the same sample by centrifugation. The following whole blood (WB) to plasma (P) ratios were used: 1WB:0P (undiluted), 2WB:1P, 1WB:1P. Than *S. Aureus*-GFP was added to all samples with a MOI of 1 (bacteria to phagocytes ratio of 1:1). The whole blood with bacteria was shaken at 37°C for 40 minutes. After 20 and 40 minutes, part of the sample was removed and put on ice. Leukocytes were stained for 15 minutes on ice with CD45-APC for recognition during analysis. After staining, cells were fixed with paraformaldehyde (PFA) 1% and the percentage of GFP-positive neutrophils was measured on the BD LSRFortessa™. Data are presented as mean. The percentage of GFP-positive neutrophils was compared between samples with different dilutions using the Friedman test for paired data. Dilution of whole blood led to a decrease in the percentage of GFP-positive neutrophils. This decrease was significant in samples analyzed after 20 minutes of incubation (p = 0.001). GFP = Green fluorescent protein. MOI = multiplicity of infection. *S. Aureus* = *Staphylococcus Aureus*.
(PDF)

**S1 Table. Correlation between neutrophil marker expression and white cell viability fraction.**
(DOCX)

**S1 File.**
(PDF)

## Acknowledgments

The authors would like to thank Maarten ten Berg, Mark de Groot and Imo Höfer for their help with data extraction. Also, the authors would like to thank Nienke Vrisekoop for her insightful suggestions and her help with experiments. No funding, equipment or other materials were received for this study.

## Author Contributions

**Conceptualization:** Lillian Hesselink, Roy Spijkerman, Pien Hellebrekers, Karlijn J. P. Van Wessem, Leo Koenderman, Luke P. H. Leenen, Falco Hietbrink.

**Data curation:** Lillian Hesselink, Roy Spijkerman, Robert J. van Bourgondiën, Enja Blasse, Saskia Haitjema, Albert Huisman.

**Formal analysis:** Lillian Hesselink, Karlijn J. P. Van Wessem.

**Investigation:** Lillian Hesselink, Roy Spijkerman, Pien Hellebrekers, Robert J. van Bourgondiën, Enja Blasse, Saskia Haitjema, Albert Huisman, Luke P. H. Leenen, Falco Hietbrink.

**Methodology:** Lillian Hesselink, Karlijn J. P. Van Wessem, Falco Hietbrink.

**Project administration:** Lillian Hesselink.

**Resources:** Wouter W. van Solinge, Leo Koenderman, Luke P. H. Leenen.

**Supervision:** Pien Hellebrekers, Wouter W. van Solinge, Leo Koenderman, Luke P. H. Leenen, Falco Hietbrink.

**Validation:** Lillian Hesselink.

**Visualization:** Lillian Hesselink, Roy Spijkerman.

**Writing – original draft:** Lillian Hesselink.

**Writing – review & editing:** Roy Spijkerman, Pien Hellebrekers, Robert J. van Bourgondiën, Enja Blasse, Saskia Haitjema, Albert Huisman, Wouter W. van Solinge, Karlijn J. P. Van Wessem, Leo Koenderman, Luke P. H. Leenen, Falco Hietbrink.

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
