## [Decision Letter · Decision Letter 0]

30 Apr 2020

PONE-D-20-09054

Fragile neutrophils in surgical patients: a phenomenon associated with critical illness.

PLOS ONE

Dear Dr. Hesselink,

Thank you for submitting your manuscript to PLOS ONE. After careful consideration, we feel that it has merit but does not fully meet PLOS ONE’s publication criteria as it currently stands. Therefore, we invite you to submit a revised version of the manuscript that addresses the points raised during the review process. Both external reviewers gave specific comments. Provide in your revised version a point-by-point reply to these comments.

We would appreciate receiving your revised manuscript by Jun 14 2020 11:59PM. To enhance the reproducibility of your results, we recommend that if applicable you deposit your laboratory protocols in protocols.io, where a protocol can be assigned its own identifier (DOI) such that it can be cited independently in the future. For instructions see: http://journals.plos.org/plosone/s/submission-guidelines#loc-laboratory-protocols

We look forward to receiving your revised manuscript.

Kind regards,

Paul Proost, Ph.D.

Academic Editor

PLOS ONE

Reviewers' comments:

Reviewer's Responses to Questions

**Comments to the Author**

1. Is the manuscript technically sound, and do the data support the conclusions?

Reviewer #1: Partly

Reviewer #2: Yes

2. Has the statistical analysis been performed appropriately and rigorously? 

Reviewer #1: Yes

Reviewer #2: Yes

3. Have the authors made all data underlying the findings in their manuscript fully available?

Reviewer #1: Yes

Reviewer #2: Yes

4. Is the manuscript presented in an intelligible fashion and written in standard English?

Reviewer #1: Yes

Reviewer #2: Yes

5. Review Comments to the Author

Reviewer #1: Interesting study describing the presence of a neutrophil subset with enhanced vulnerability that may occur in the circulation of critically ill surgical patients. The presence of these cells potentially holds valuable information concerning the clinical condition of patients.

Specific comments:

1) Please indicate the relative abundance of fragile neutrophils (percentage of total neutrophil subset) for patients that have these particular cells.

2) Data shown in figure 3 provide evidence for alterations in membrane marker expression of patient and control neutrophils. The observed differences might be related to differences in pathological state/inflammatory environment (e.g. alterations in plasma cytokine and chemokine levels that may cause neutrophil phenotypical and functional differences) rather than the presence of fragile neutrophils. Results from patients that do not have fragile neutrophils should be included in figures 3 and 4.

3) Did the authors evaluate the plasma cytokine/chemokine levels in patients?

4) Is there a correlation between relative abundance of fragile neutrophils and alterations in membrane marker expression?

5) Data are too preliminary to make general conclusions concerning (absence of) alterations in neutrophil functionality in patients with fragile neutrophils. Assays were performed using whole blood samples or total leukocytes and only assess fMLF responsiveness and phagocytosis.

6) Authors indicate that samples were processed within 3.5 hours, which can be tricky when working with neutrophils. Is there a correlation between time to processing and number of fragile neutrophils/phenotypical differences?

7) Did the authors ever detect fragile neutrophils in control samples?

8) Is there any evidence for morphological differences (e.g. differences in size or nuclear segmentation properties) of the fragile neutrophil subset?

9) Typo in Figure 3 panel A

Reviewer #2: In the study entitle "Fragile neutrophils in surgical patients: a phenomenon associated with critical illness", the authors identified a small fraction of circulating neutrophils from surgical patients that was correlated to adverse clinical outcomes. Although there has been no deep investigation about the origin and function of these cells specially in infectious diseases, preliminary analysis were performed to characterize surface markers, opening an important window for further investigations.

Minor comment;

- Figure 3: The authors mentioned in the text that expression of CBRM1/5 and CD14 were slightly higher in patient samples incubated fMLF. However, based on Figure 3J, CD14 expression is reduced in fMLF-stimulated neutrophils from patients compared to those from healthy controls.

6. PLOS authors have the option to publish the peer review history of their article (what does this mean?). If published, this will include your full peer review and any attached files.

Reviewer #1: No

Reviewer #2: Yes: Flavio Almeida Amaral

---

## [Author Response · Author response to Decision Letter 0]

14 Jun 2020

Reviewer #1

Thank you for reviewing our manuscript and your compliments on this study. With your suggestions we added a few essential explanations and extra data to the manuscript. 

Comments:

1) Please indicate the relative abundance of fragile neutrophils (percentage of total neutrophil subset) for patients that have these particular cells.

Thank you for this suggestion. We added this to table 1 (page 12). Also, Fig S2 gives insight in the number of non-viable leukocytes and the total number of leukocytes over time. 

2) Data shown in figure 3 provide evidence for alterations in membrane marker expression of patient and control neutrophils. The observed differences might be related to differences in pathological state/inflammatory environment (e.g. alterations in plasma cytokine and chemokine levels that may cause neutrophil phenotypical and functional differences) rather than the presence of fragile neutrophils. Results from patients that do not have fragile neutrophils should be included in figures 3 and 4.

We completely agree that the differences in membrane marker expression can be related to differences in the inflammatory environment, such as alterations in plasma cytokines and chemokine levels, and that they are not the result of fragile neutrophils per se. We obtained these patient data, in combination with data from healthy controls, to get insight in the phenotypic characteristics of the complete neutrophil compartment of patients with fragile neutrophils. Although we were not able to include patients without fragile neutrophils in the prospective study, we have published extensively on the changes of neutrophils during acute systemic inflammation without the presence of fragile neutrophils. We have added these studies and considerations to the discussion section, paragraph 3 and 4.

3) Did the authors evaluate the plasma cytokine/chemokine levels in patients?

We did not evaluate plasma cytokine/chemokines levels in these patients. We agree that plasma cytokine/chemokines levels can give more insight into the inflammatory response. However, these mediators are divers and numerous and there is no real consensus which of these are important drivers of neutrophil inflammation. We chose to focus on the effector cell, the neutrophil, because neutrophils integrate these signals into a cellular response. As such, we hypothesized that neutrophil responses can be studied as “simple” biomarker of a complex inflammatory response. For this proof-of-principle study, we aimed to identify the presence of these fragile neutrophils and describe the phenomenon as such. Due to the very low incidence, inclusion and analysis is challenging. Nevertheless, in future studies, a more extensive investigation of other inflammatory mediators might be valuable. We have added these considerations to the discussion (line 306). 

4) Is there a correlation between relative abundance of fragile neutrophils and alterations in membrane marker expression?

We analyzed the correlation between the white cell viability fraction (WVF) versus neutrophil marker expression in the prospectively included patients with fragile neutrophils (n = 9). There was no significant correlation between neutrophil marker expression and white cell viability fraction. This was added to the fourth paragraph of the discussion (line 271) and added as Supplementary Table S4. 

We cannot rule out that no significant correlation was found due to the small study group we investigated, however, no marker stood out. Future studies should investigate neutrophil phenotypes in a larger cohort of patients with fragile neutrophils. We added this suggestion to the discussion, paragraph 6, line 306. 

WVF vs. Pearson correlation, R P-value

CD35 0.432 0.246

CD66b 0.522 0.150

CD64 -0.038 0.922

CBRM1/5 0.364 0.336

CD11b 0.521 0.150

CD14 -0.016 0.967

CD16 0.278 0.469

CD62L -0.163 0.675

LAIR1 0.184 0.636

CD49d 0.559 0.249

5) Data are too preliminary to make general conclusions concerning (absence of) alterations in neutrophil functionality in patients with fragile neutrophils. Assays were performed using whole blood samples or total leukocytes and only assess fMLF responsiveness and phagocytosis.

We agree that data on neutrophil functionality is still preliminary. Therefore we added the following sentences to the discussion: 

Line 277: “These preliminary data on neutrophil functionality (responsiveness to fMLF and phagocytosis of bacteria) suggest that neutrophil function was not impaired in the prospectively included patients with a decreased WVF.”

Line 308: Due to this, data on neutrophil functionality in these patients is still preliminary. Future research should focus on other functional neutrophil tests to gain better insight in how the presence of fragile neutrophils influences overall neutrophil functionality.

6) Authors indicate that samples were processed within 3.5 hours, which can be tricky when working with neutrophils. Is there a correlation between time to processing and number of fragile neutrophils/phenotypical differences?

Blood sampling was always performed within 3.5 hours after the viability alarm went off (line 408). With that we meant that within 3.5 hours after the alarm, we obtained a new blood sample from the patient that was analyzed immediately. To check if the decreased white cell viability fraction (WVF) was not a result of processing time and to check if the fragile cells were still present in the peripheral blood of the patient, this new blood sample was again tested for the presence of fragile neutrophils by the hematology analyzer. We used this quick and fully automated hematology analyzer (Cell-dyn Sapphire) to minimize the time effect and increase the reproducibility of the test. This test was always performed within 15 minutes. Afterwards we started the sample processing, so that there was basically no time delay between the blood drawing and the start of the sample work-up. In all patients, fragile neutrophils were still present in the second blood sample. 

To clarify this, we added this explanation to the methods section, line 410. 

7) Did the authors ever detect fragile neutrophils in control samples?

Fragile neutrophils were never detected in control samples that were processed within 15 min. With a delay of 3-6h, no more than 2% fragile neutrophils were found in control samples. 

8) Is there any evidence for morphological differences (e.g. differences in size or nuclear segmentation properties) of the fragile neutrophil subset?

If the necessary laboratory equipment was available, we also performed cell sorting in which we gated the PI+ (non-viable) cells and the PI- cells and made a cytospin. See the Figures below. Below are a few examples of the many cytospins we looked at. We analyzed these cytospins using the fluorescence microscope and found that the PI+ fragile neutrophils did not stay intact during sorting (black circles are examples in Figures 1 and 2. For figures, see attached file). The cells that were not destroyed in de sort were mostly PI negative and presumably ended up in the PI+ sort because they stuck to PI+ neutrophils (red circles are examples in Figures 1 and 2. For figures, see attached file). Hence, we were technically not able to find specific characteristics of the fragile neutrophil subset. 

However, we did find some remarkable morphological characteristics in cytospins of patients with fragile neutrophils (in cells that stayed intact). Firstly, often these patients had many banded neutrophils and progenitor cells. Secondly, we found cytoplasmic vacuoles and toxic granulation in these neutrophils, which are all signs of severe infections/inflammation.[2,3] These findings are in line with other findings that suggest severe inflammation in patients with fragile neutrophils, such as the changes in marker expression found in these patients and the number of severe infections found in these patients.

We added this to the discussion, line 259, and added two examples as supplement (S3 Fig). 

9) Typo in Figure 3 panel A

Thank you for noticing this typo. We changed this in the Figure.

 

Reviewer #2

Thank you for your time and effort in reviewing the manuscript. Also thank you for your compliments regarding the importance of the paper. 

Comments:

1. Figure 3: The authors mentioned in the text that expression of CBRM1/5 and CD14 were slightly higher in patient samples incubated fMLF. However, based on Figure 3J, CD14 expression is reduced in fMLF-stimulated neutrophils from patients compared to those from healthy controls.

Thank you for noticing this mistake. Indeed, CD14 expression was lower. We adjusted this in the text: “CBRM1/5 was slightly higher (p = 0.034) and CD14 was slightly lower (p = 0.023) than in healthy controls. “ (line 141).

References 

1 Kabutomuri O. Toxic Granulation Neutrophils and C-Reactive Protein. Arch Intern Med. 2000;160(21):3326–3327.

2 McCall CE, Katayama I, Cotran RS, Finland M. Lysosomal and ultrastructural changes in human "toxic" neutrophils during bacterial infection. J Exp Med. 1969;129(2):267‐293. doi:10.1084/jem.129.2.267

---

## [Decision Letter · Decision Letter 1]

1 Jul 2020

PONE-D-20-09054R1

Fragile neutrophils in surgical patients: a phenomenon associated with critical illness.

PLOS ONE

Dear Dr. Hesselink,

Thank you for submitting your manuscript to PLOS ONE. After careful consideration, we feel that it may be accepted upon minor revision. Could you comment on the medication of the patient population in view of the large vacuoles in the cells in your new figure S3 and the reviewer's comment on corticosteroids being associated with vacuoles in neutrophils.

We look forward to receiving your revised manuscript.

Kind regards,

Paul Proost, Ph.D.

Academic Editor

PLOS ONE

Reviewers' comments:

Reviewer's Responses to Questions

**Comments to the Author**

1. If the authors have adequately addressed your comments raised in a previous round of review and you feel that this manuscript is now acceptable for publication, you may indicate that here to bypass the “Comments to the Author” section, enter your conflict of interest statement in the “Confidential to Editor” section, and submit your "Accept" recommendation.

Reviewer #1: All comments have been addressed

Reviewer #2: All comments have been addressed

2. Is the manuscript technically sound, and do the data support the conclusions?

Reviewer #1: Yes

Reviewer #2: Yes

3. Has the statistical analysis been performed appropriately and rigorously? 

Reviewer #1: Yes

Reviewer #2: Yes

4. Have the authors made all data underlying the findings in their manuscript fully available?

Reviewer #1: Yes

Reviewer #2: Yes

5. Is the manuscript presented in an intelligible fashion and written in standard English?

Reviewer #1: Yes

Reviewer #2: Yes

6. Review Comments to the Author

Reviewer #1: Thank you for responding so positively to my comments. I think the article has been improved by the review process and I hope you agree.

One last remark regarding the vacuoles (suppl. fig. S3): we usually observe these kind of 'vacuolated' neutrophils in patients that received corticosteroids. It may be interesting to check whether there is an association with therapy.

Reviewer #2: The authors have answered the raised question, making a correction in the result description. Although this is preliminary study, with first analysis to characterize surface markers of fragile neutrophils, this study opens an important window for further investigations on origin and functions of this neutrophil subtype.

7. PLOS authors have the option to publish the peer review history of their article (what does this mean?). If published, this will include your full peer review and any attached files.

Reviewer #1: No

Reviewer #2: **Yes: **Flavio Almeida Amaral

---

## [Author Response · Author response to Decision Letter 1]

7 Jul 2020

Thank you again for reviewing our manuscript and reviewing the responses. Also, thank you for the insightful suggestions that improved the manuscript. 

Comment: “One last remark regarding the vacuoles (suppl. fig. S3): we usually observe these kind of 'vacuolated' neutrophils in patients that received corticosteroids. It may be interesting to check whether there is an association with therapy.”

The prospectively included patients who had these vacuolated neutrophils did not receive corticosteroids before or during blood sampling. Because this was a relatively small group of patients, we additionally analyzed systemic corticosteroid use in the retrospectively included patients with fragile neutrophils. We found that 11 of the 74 patients (14.9%) were administered corticosteroids during admission. However, only 2 patients received corticosteroids during blood sampling. Moreover, no consistency in time-dependence could be observed between the start of corticosteroids and the presence of fragile neutrophils in the blood stream. Hence, it seems unlikely that the administration of corticosteroids was related to the presence of neutrophils with cytoplasmic vacuoles in patients with fragile neutrophils.

We added the following text to the methods section, line 536: “Also, it was investigated whether there was a relation between administration of corticosteroids and a decreased WVF, since corticosteroids are known to interfere with apoptosis of immune cells[1, 2]”

The following text was added to the results section, line 195: “Also, patients with fragile neutrophils more often received corticosteroids during admission (7.6% versus 14.9%, p = 0.027). However, only two (2.7%) patients received corticosteroids at the moment the viability alarm went off and no time-dependence was observed between the start of corticosteroids and the day the alarm went off.”

We added the following text to the discussion, line 265: “These characteristics are typical for severe inflammation and/or infection[3, 4] and can sometimes be seen after administration of corticosteroids[5]. However, the prospectively included patients who displayed these characteristics did not receive corticosteroids.”

References

1. Schmidt S, Rainer J, Ploner C, et al (2004) Glucocorticoid-induced apoptosis and glucocorticoid resistance: Molecular mechanisms and clinical relevance. Cell Death Differ 11:S45–S55. https://doi.org/10.1038/sj.cdd.4401456

2. Gruver-Yates A, Cidlowski J (2013) Tissue-Specific Actions of Glucocorticoids on Apoptosis: A Double-Edged Sword. Cells 2:202–223. https://doi.org/10.3390/cells2020202

3. Kabutomkori O, Iwatani Y, Kanakura Y (2000) Toxic granulation neutrophils and C-reactive protein. Arch Intern Med 160:3326–7. https://doi.org/10.1001/archinte.160.21.3326-a

4. McCall CE, Katayama I, Cotran RS, Finland M (1969) Lysosomal and ultrastructural changes in human “toxic” neutrophils during bacterial infection. J exp med 129:267–293

5. Lee S, Khankhanian P, Mascarenhas JO (2015) Corticosteroid-induced morphological changes in cells of the myeloid lineage. Am J Hematol 90:679–680. https://doi.org/10.1002/ajh.23943

---

## [Editor Report · Decision Letter 2]

10 Jul 2020

Fragile neutrophils in surgical patients: a phenomenon associated with critical illness.

PONE-D-20-09054R2

Dear Dr. Hesselink,

We’re pleased to inform you that your manuscript has been judged scientifically suitable for publication and will be formally accepted for publication once it meets all outstanding technical requirements.

Kind regards,

Paul Proost, Ph.D.

Academic Editor

PLOS ONE
---

## [Editor Report · Acceptance letter]

22 Jul 2020

PONE-D-20-09054R2 

Fragile neutrophils in surgical patients: a phenomenon associated with critical illness. 

Dear Dr. Hesselink:

I'm pleased to inform you that your manuscript has been deemed suitable for publication in PLOS ONE. Congratulations! Your manuscript is now with our production department. 

Kind regards, 

on behalf of

Dr. Paul Proost 

Academic Editor

PLOS ONE